# Ancestral male recombination in *Drosophila albomicans* produced geographically restricted neo-Y chromosome haplotypes varying in age and onset of decay

**Kevin H-C. Wei** (ID)**, Doris Bachtrog** (ID)*

Department of Integrative Biology, University of California Berkeley, Berkeley, California, United States of America

* dbachtrog@berkeley.edu

**Data Availability Statement:** All raw sequences can be found under BioProject ID PRJNA554143. Relevant scripts used for this study can be found at KW's github page, https://github.com/weikevinhc/

## Abstract

Male Drosophila typically have achiasmatic meiosis, and fusions between autosomes and the Y chromosome have repeatedly created non-recombining neo-Y chromosomes that degenerate. Intriguingly, *Drosophila nasuta* males recombine, but their close relative *D. albomicans* reverted back to achiasmy after evolving neo-sex chromosomes. Here we use genome-wide polymorphism data to reconstruct the complex evolutionary history of neo-sex chromosomes in *D. albomicans* and examine the effect of recombination and its cessation on the initiation of neo-Y decay. Population and phylogenomic analyses reveal three distinct neo-Y types that are geographically restricted. Due to ancestral recombination with the neo-X, overall nucleotide diversity on the neo-Y is similar to the neo-X but severely reduced within neo-Y types. Consistently, the neo-Y chromosomes fail to form a monophyletic clade in sliding window trees outside of the region proximal to the fusion. Based on tree topology changes, we inferred the recombination breakpoints that produced haplotypes specific to each neo-Y type. We show that recombination became suppressed at different time points for the different neo-Y haplotypes. Haplotype age correlates with onset of neo-Y decay, and older neo-Y haplotypes show more fixed gene disruption via frameshift indels and down-regulation of neo-Y alleles. Genes are downregulated independently on the different neo-Ys, but are depleted of testes-expressed genes across all haplotypes. This indicates that genes important for male function are initially shielded from degeneration. Our results offer a time course of the early progression of Y chromosome evolution, showing how the suppression of recombination, through the reversal to achiasmy in *D. albomicans* males, initiates the process of degeneration.

## Author summary

Sex chromosomes evolve from autosomes yet harbor many unique characteristics that differentiate them from autosomes and from each other. Male-specific Y-chromosomes degenerate: they lose most of their protein-coding genes and accumulate repetitive DNA.

albneoY. Tree files and vcf files can be found on Dryad at https://datadryad.org/stash/dataset/doi:10.6078/D12T2X.

**Funding:** This research was supported by NIH grants (R01GM076007, R01GM101255 and R01GM093182) to DB. The funders had no role in study design, data collection and analysis, decision to publish, or preparation of the manuscript.

**Competing interests:** The authors have declared that no competing interests exist.

How Y chromosomes evolve these unique features is poorly understood, especially during the initial stages of Y decay. Here, we use genome-wide polymorphism data to study the unusual neo-sex chromosomes of *D. albomicans*. Its neo-Y is the youngest neo-sex chromosome analyzed to date, which allows us to examine the molecular changes and evolutionary processes initiating Y degeneration. We find both coding and non-coding changes contributing to early neo-Y decay, and genes on the neo-Y are starting to become downregulated. Genes with testis-biased expression are more likely to be preserved on the neo-Y, indicating that genes important for male function are shielded from degeneration early on. Most intriguingly, we find that this sex chromosome formed in an ancestor where males (and thus the neo-Y) were recombining, and males reverted back to achiasmy only some time after the establishment of the neo-sex chromosomes. This resulted in several unusual patterns of neo-Y evolution, and allowed us to study the complex interplay of recombination, selection and population structure on neo-Y decay.

## Introduction

Sex chromosomes originate from a pair of homologous autosomes, yet they are often highly differentiated in morphology and function [1]. This transformation starts when a sex-determining factor arises on an autosome causing it to have sex-specific transmission. Sexually antagonistic mutations, which are variants that are beneficial to one sex but detrimental to the other, are expected to accumulate in close proximity to the sex-determining factor [2,3]. To ensure linkage of the sex-determining factor with sexually antagonistic alleles, recombination between the nascent X and Y chromosomes will become suppressed [2,3]. This can happen in a step-wise fashion, where recombination suppression first occurs around the sex-determining locus. Subsequent gains of sex-beneficial alleles across the chromosome can lead to the expansion of the non-recombining region (for review see [4]). Such a stepwise process of recombination suppression has been documented in multiple taxa, including mammals [5,6], birds [7,8], fish [9], or plants [10,11].

The cessation of recombination between the sex chromosomes marks a pivotal step that leads the X and Y chromosome down a predictable trajectory of differentiation. The absence of recombination on the Y implies that mutations on the same chromosome cannot become unlinked by crossing-over [2,12,13]. Interference among linked mutations can lead to the irreversible accumulation of deleterious mutations on the Y, and in the long term, the degeneration and subsequent loss of all of its ancestral genes [1,3,13,14]. In particular, positive selection for an advantageous variant on the Y will cause a sweep of the entire chromosome, dragging along all linked deleterious alleles to fixation [14,15]. Similarly, negative selection on the Y chromosome can greatly reduce its effective population size below its neutral expectation of 1/4 of the effective population size of autosomes, thereby increasing the rate of fixation of weakly deleterious alleles [13,16]. Finally, Y chromosomes may irreversibly accumulate deleterious mutations by a process known as Muller's ratchet, where Y chromosomes with the fewest number of deleterious mutations are lost by genetic drift [2,17]. All these processes lead to an accumulation of deleterious mutations on the Y, and greatly reduce levels of nucleotide diversity [13,16]. Over long evolutionary time periods, this often leads to a degenerate Y chromosome replete with pseudogenes and repetitive elements, and the Y is often transcriptionally suppressed by heterochromatin formation [18–28]. During the gradual loss of genes on the Y, the X acquires mechanisms of dosage compensation to balance reduced expression of sex-linked genes [29]. In Drosophila males (XY), this is accomplished by chromosome-wide up-

regulation of X-linked genes by epigenetic mechanisms, and a similar mechanism convergently evolved in green anole lizards [30]. However, dosage compensation can also be achieved through gene-by-gene up-regulation as exemplified by genes on the Z chromosome of female (ZW) birds [31,32] or snakes [33].

Unlike most sexually reproduced organisms, *Drosophila* is peculiar since males are achiasmatic–that is, recombination does not occur during male meiosis. Therefore, autosomes that become fused to the ancestral Y chromosome (so-called neo-Y chromosomes) will be transmitted through males only. Neo-Y chromosomes will immediately stop any genetic exchange with their former homologs (termed neo-X chromosomes), thereby bypassing the stepwise formation of non-recombining regions. Neo-sex chromosomes have formed independently multiple times in the Drosophila genus by fusions of autosomes (referred to as Muller elements [34,35]) to the ancestral sex chromosomes (Muller A) [36]. Neo-Ys at various state of degeneration within the Drosophila genus reflect their different time of origination. Among the oldest is the neo-Y of *D. pseudoobscura*, which is likely the remnant of Muller D [37,38]. Its autosomal origin is nearly unrecognizable after 15 million years of degeneration, with very few genes remaining on the neo-Y [37,38]. Intermediate levels of degeneration can be found on the neo-Y of *D. miranda* which arose ~1.5 million years ago through the fusion of Muller C to the Y [39]. Over 90% of the ancestral protein coding genes can still be identified on the neo-Y, but 40% have become non-functional through frameshifts, nonsense mutations, deletions and transposable element insertions [40–43]. On the younger neo-Y of *D. busckii*, which formed through the fusion of Muller F with the sex chromosomes around 0.8 million years ago, over 60% of genes have become pseudogenized [44,45]. Despite the different ages, heavy enrichment of repressive chromatin marks can be found on all these neo-Ys [30,32].

One of the youngest known neo-sex chromosome pairs is found in *D. albomicans*, a species distributed primarily across the Asia Pacific island chain (Fig 1) [46,47]. *D. albomicans* is a young species that recently diverged from the West African species *D. nasuta* between 150 and 500 thousand years ago [48,49]. These otherwise indistinguishable sister species produce fully fertile hybrids [50,51], but differ karyotypically by two Robertsonian fusions. Chromosome 3 (henceforth Chr. 3), which is itself a fusion between Muller C and D common to all species of the *nasuta* clade fused to both the ancestral X and Y [52,53]. These two fusions produced two massive neo-sex chromosomes in *D. albomicans* that are each over 55 Mb and harbor >5000 genes (Fig 1A). Previous studies have found very few coding differences between the neo-sex chromosomes [41], but the neo-Y nonetheless shows signs of reduced expression suggesting that degeneration might have begun with disruption of regulatory elements instead of coding sequences [54].

Non-recombining neo-Y chromosomes are expected to harbor low levels of sequence polymorphism [39,55]. Intriguingly, a recent study investigating levels of sequence variability at 27 neo-sex linked loci found unexpectedly high levels of polymorphism on the neo-Y of *D. albomicans*, and also high levels of shared variation with the neo-X chromosome [56]. These findings are inconsistent with either a single origin of the neo-Y, or with recombination being absent between the neo-sex chromosomes. Indeed, a recent study showed that, surprisingly, recombination occurs in *D. nasuta* males, yet is currently absent in *D. albomicans* males [56]. Thus, this suggests that ancestral recombination between the neo-sex chromosomes in *D. albomicans* males has resulted in the pattern of shared polymorphisms between them.

Together, these observations imply that the young neo-sex chromosomes of *D. albomicans* not only present a window to examine the initiating stages of sex chromosome differentiation, but also provide a unique opportunity to understand how the presence and absence of recombination affects and interacts with the process of Y degeneration. Here, we investigate patterns of neo-Y variation from six populations across the species range with whole genome sequencing (Fig 1B). Using population and phylogenetic analyses, we confirm that the neo-Y fusion

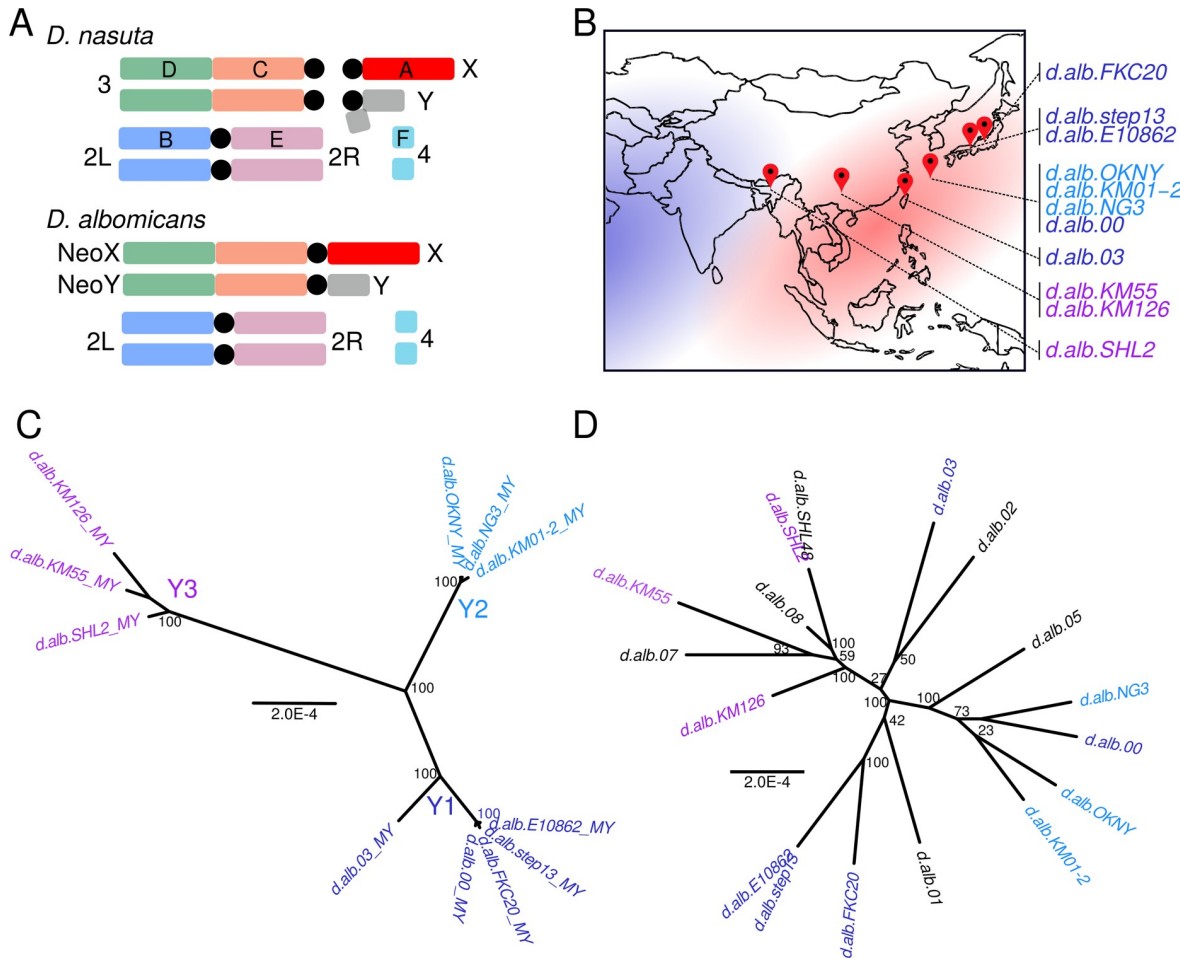

**Fig 1. The karyotypes and species range of *D. albomicans* and *D. nasuta*.** A. The male karyotypes of *D. albomicans* and *D. nasuta*. Chromosomes and their corresponding Muller elements are labeled and color-coded. Black circle denotes the centromere. B. The species ranges of *D. albomicans* and *D. nasuta* are in red and blue, respectively. Red pins denote the sources of the male strains used in this study; neo-Y samples collected from each site are labeled, and color-coded based on grouping in C. C. Unrooted maximum likelihood trees for neo-Y genotypes, based on all filtered variant sites across the chromosome, show three distinct neo-Y groups, which are labeled Y1, Y2, and Y3, and color-coded blue, light blue, and purple, respectively. Unless otherwise stated, lines from the different Y groups will maintain this color coding. D. Same as C, but for the neo-X genotypes; lines are color labeled by their neo-Y grouping, if available. The neo-X tree with only individuals with their neo-Ys genotyped (individuals in 1C) can be found in S1 Fig.

occurred once, and recombination with the neo-X generated multiple neo-Y haplotypes that are geographically restricted. Curiously, the Chr. 3 that fused with the ancestral Y carried a haplotype now only found in *D. nasuta*, which was subsequently broken down by male recombination. We show that different Y-haplotypes have lost recombination at different time points in the past, and their age is associated with different extents of degeneration. These results untangle the complex series of events during the formation and subsequent evolution of the neo-Y in *D. albomicans* and provide timed snapshots of the effect cessation of recombination has on the beginning of sex chromosome differentiation.

## Results

### Genomic patterns of neo-sex chromosome variation

In order to characterize the geographic distribution of neo-Y chromosomes, we selected eleven inbred lab strains of *D. albomicans* that were collected from Shilong (India), Kunming

(China), Nankung (Taiwan), Okinawa, Kyoto, and Fukui (Japan; S1 Table). Males and females of each strain were genotyped using Illumina high throughput sequencing. Since the current reference is a female assembly lacking the neo-Y chromosome, neo-Y genotypes were determined from neo-X sites that are heterozygous in males but homozygous in females; the neo-Y alleles are the SNPs absent in females. This generated 255,010 variant neo-Y sites across the eleven neo-Ys and 584,333 variant sites on both neo-sex chromosomes across all 28 strains after filtering. We validated 14 out of 14 randomly chosen neo-Y alleles that coincided with RFLP sites, confirming the efficacy of our approach (S2 Table). Application of the same bioinformatics pipeline on an autosome identified very few male-specific SNPs confirming that our approach is highly sensitive to detect neo-Y specific SNPs (S3 Table). Using all sites across the neo-Y chromosome, we constructed a maximum likelihood tree that revealed three distinct groups of neo-Ys (Fig 1C). Group 1 and group 2, henceforth Y1 and Y2, include flies from Taiwan and Japan, with Y2 composed exclusively of flies from the Okinawa Island. The Shilong and Kunming lines form a distinct and distant group, henceforth Y3. The three groups are separated by long branches, but individuals are highly similar within groups. This starkly contrasts to the phylogeny of the neo-X chromosome, where branches lack strong clustering and are substantially longer between individuals, indicative of higher rates of polymorphisms and less differentiation between neo-X chromosomes (Fig 1D, S1 Fig).

## Complex patterns of diversity on the neo-Y are inconsistent with single origin or lack of recombination

The formation of a Drosophila neo-Y chromosome is typically accompanied by a reduction of effective population size and nucleotide diversity ($\pi$). Because the fusion that formed the neo-Y likely happened only once and recently [53], $\pi$ is expected to be minimal due to severe bottlenecking associated with the fixation of a single fused chromosome. While diversity (as measured by $\pi$) is low in the region proximal to the centromere, $\pi$ across the rest of the neo-Y (gray in Fig 2A) is only marginally lower than $\pi$ of the neo-X (red in Fig 2A). This is counter to expectation of diversity being completely wiped out when a single neo-Y sweeps to fixation due to a lack of recombination in Drosophila males, and differs from diversity patterns observed on other Drosophila neo-Y chromosomes [4]. Interestingly, within each Y group, we find large swaths of the chromosomes with minimal $\pi$, indicating that the Y chromosomes within a group have very low levels of polymorphism (blue (Y1), light blue (Y2), and purple (Y3) in Fig 2A). Therefore, the elevated $\pi$ across all neo-Ys (gray in Fig 2A) is the result of high differentiation among neo-Y chromosome groups. This is further supported by high levels of $F_{ST}$ between the Y groups (S2 Fig). For all the Y groups, differentiation from the neo-X is high with $F_{ST} > 0.6$ in regions proximal to the centromere, followed by precipitous drops to ~0.4 (Fig 2B, S3 Fig, S4 Fig). For Y1 (red) and Y2 (green), the drops occur more distally from the centromere at 50.5 Mb than for Y3 (blue) at 53.2 Mb. Y1 and Y2 both contain additional valleys of $F_{ST}$ spanning large parts of the chromosome arm. The drops in $F_{ST}$ are primarily due to increases in shared polymorphism between the neo-X and neo-Y along the chromosome (Fig 2C). The presence of multiple Y groups, varying levels of differentiation and shared polymorphism between the neo-X and neo-Y are inconsistent with a simple model of neo-Y origination in a non-recombining background. The centromere proximal region and the telomeric end are the only regions with the expected pattern of high proportion of fixed differences (Fig 2C).

To better understand what causes these variable patterns of differentiation, we generated maximum likelihood trees for the neo-Xs and neo-Ys in 200kb sliding windows across the chromosome. We find that the centromere proximal region shows the expected topology,

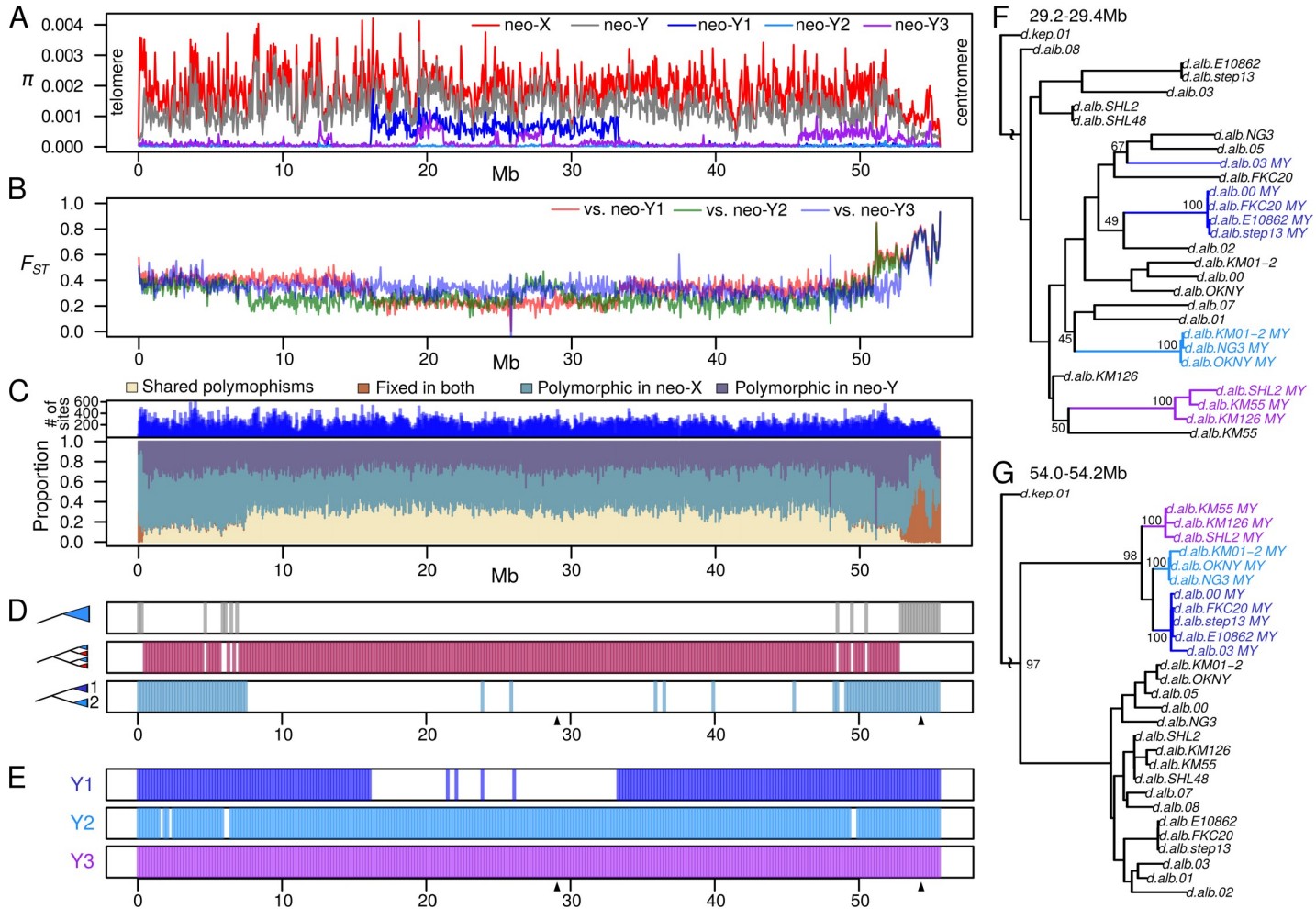

**Fig 2. Recombination created distinct neo-Y groups after the Y-autosome fusion.** A. Nucleotide diversity estimates for the neo-X, all neo-Ys, and neo-Y groups across the chromosome are plotted in 50kb non-overlapping windows. B. Population differentiation estimates, $F_{ST}$, between the neo-X and the three Y groups. See S2 Fig for $F_{ST}$ plotted individually. C. The proportion of shared polymorphism and fixed differences between the neo-X and neo-Y in 50kb non-overlapping windows across the chromosome is depicted; singletons are removed, and only biallelic sites are used. The number of informative variant sites (used as the denominator) is plotted in the top track. D. Based on 200kb sliding window maximum likelihood trees of the neo-Xs and neo-Ys, three different topologies are plotted: windows where all the neo-Ys form a monophyletic group (top track), windows where the neo-Ys are paraphyletic (middle track), windows where Y1 and Y2 individuals are monophyletic (bottom track). Arrow heads underneath the tracks denote windows where the examples (F & G) are taken from. E. Same as D, but shows windows where individuals within the same Y type are monophyletic. F-G. Trees at windows that exemplify the different topologies; lines from the different Y groups are color coded as in Fig 1C. Bootstrap support values are displayed for the neo-X and neo-Y split nodes, and Y group nodes.

where all the neo-Y chromosomes form a monophyletic clade with a deep split from all the neo-Xs (Fig 2D and 2G). Monophyly of neo-Ys close to the centromere confirms that the fusion that created the neo-Y (that is, between Chr. 3 and the ancestral Y) occurred only once. However, monophyly is interrupted at 53.2Mb which coincides with the decline in differentiation between Y3 and the neo-X (Fig 2D). For the vast majority of the chromosomal windows, the neo-Ys are interspersed within the neo-Xs, instead of forming a monophyletic clade (Fig 2D and 2F). When Y3 individuals are excluded, Y1 and Y2 group together for more windows along the neo-sex chromosome (Fig 2D); the regions where they are no longer monophyletic are associated with lowered differentiation from and increased shared polymorphism with the neo-Xs (Fig 2B–2D). Notably, even though the three Y groups are paraphyletic for most of the chromosome, individuals within them are nearly always grouped together (Fig 2E), consistent

with the low within-group π. The only exception is the Taiwanese strain (d.alb 03) which is paraphyletic to the rest of the Y1 individuals for trees in the middle of the chromosome (Fig 2E and 2F). This topology change is also reflected by elevated π within Y1 (blue in Fig 2A) and a valley of $F_{ST}$ between Y1 and the neo-X (red in Fig 2B). Despite supporting the single origin of the neo-Y fusion near the centromere, the complex patterns of diversity, differentiation and tree topologies across the rest of the chromosome are incompatible with the neo-Y being genetically isolated from the neo-X due to achiasmy in Drosophila males. Instead, these results are consistent with multiple haplotypes of neo-Ys that have been introduced through recombination with the neo-X.

## Male recombination produced large neo-Y haplotype blocks of different ages

The valleys of Fst and plateaus of shared polymorphism between the neo-X and neo-Y suggest that the different neo-Ys have megabase-sized haplotype blocks. We reconstructed individual neo-Y haplotypes based on tree topologies across the chromosome, reasoning that disruption of monophyly of the neo-Ys is due to recombination events with neo-Xs (Fig 3A). For example, monophyly of the neo-Y at the centromere proximal region indicates that there is only one haplotype; this haplotype is disrupted when Y3 is no longer monophyletic with the rest of the neo-Ys, indicating a second haplotype. We find that each Y group has a unique combination of haplotypes resulting from recombination breakpoints distal to the centromere (Fig 3B). The largest haplotype block spans nearly the entirety of the chromosome from 0.4 Mb to 53.2 Mb and distinguishes Y3 from the others. A slightly smaller haplotype block between 7.6Mb and 49.6Mb distinguishes Y2 from Y1. While we initially categorized the Taiwanese strain (d. alb.03) as Y1, it harbors a different haplotype in the middle of the chromosome from 16.2 Mb to 33.2 Mb separating it from the remaining Y1 individuals. This additional haplotype also explains elevated π within group Y1 (Fig 2B). Interestingly, all the individuals have the same haplotype at both ends of the neo-Y chromosome. While monophyly at the centromeric end is expected due to the single origin of the Y-autosome fusion, a single haplotype at the telomeric regions is surprising, and may suggest selection near the telomere leading to the fixation of a particular haplotype variant.

Given that *D. albomicans* males no longer recombine [56], the number of nucleotide changes occurring along different branches of the phylogeny can be used to roughly date the age of the Y groups, and the time since cessation of recombination (see Methods for details). For each haplotype block (Fig 3B bottom), we first took the branch length from the node of the species split (*s*) to the nodes that separate the neo-Y groups from the neo-X sisters (*h*) (Table 1, Fig 3C left). Shorter branch lengths indicate that after the formation of the species, the neo-Y haplotype ceased to recombine with the neo-X earlier, and, extensively, an older neo-Y haplotype. The Y3 and Y2 groups have the shortest and longest branch lengths, respectively, suggesting that they are the oldest and youngest, respectively. Second, we calculated the branch lengths from the nodes separating the neo-Y and neo-X (*h*) to the tip of the trees, which indicate the amount of changes after the cessation of recombination (Table 1, Fig 3C, middle and right). However, since the neo-Y haplotypes likely began to experience different rates of change after recombination stopped due to a reduction in effective population size and reduced efficacy of selection, we took the branch lengths of the neo-X sisters as well (Table 1, Fig 3C middle and right). In both cases, Y3 and its corresponding neo-X sisters have the longest branch lengths and Y2 the shortest. These results consistently indicate that Y3 and Y2 are the oldest and youngest neo-Y types, respectively, with Y1 and the Taiwanese haplotypes being of intermediate age. To ensure that the different branch length estimates are not the result of

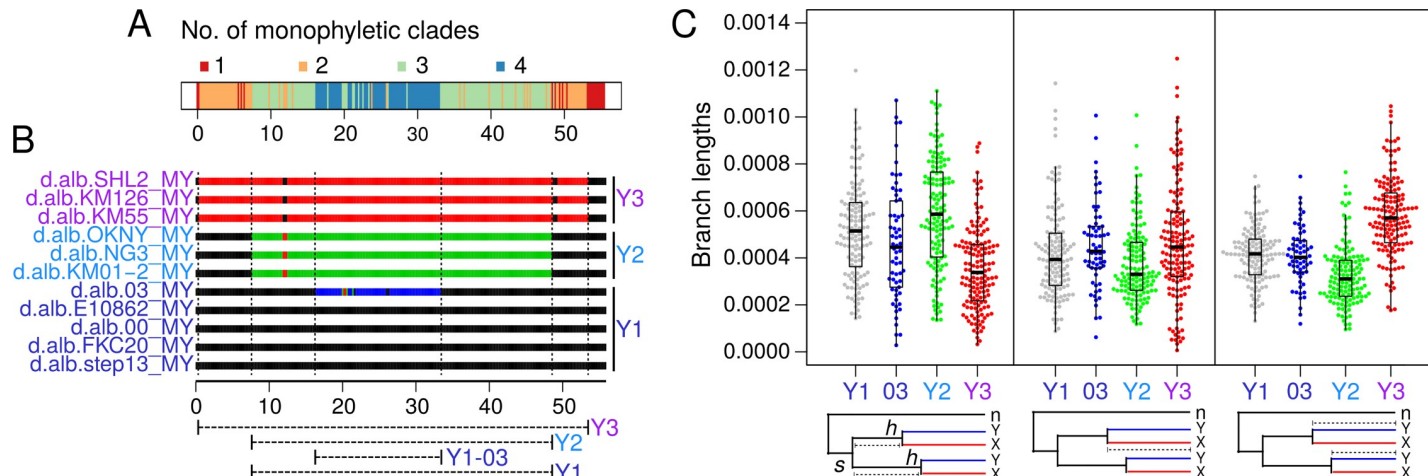

**Fig 3. Large neo-Y haplotype blocks and their ages.** A. The number of monophyletic branches to which the neo-Ys exclusively belong are inferred from the 200kb sliding window phylogenies and plotted. B. For each 200kb window of each neo-Y individual, those that belong to the same monophyletic branch, and therefore haplotypes, are color-coded the same. When single 200kb windows interrupt haplotype blocks, their colors are changed to match the flanking windows. Note the colors here are arbitrary and do not indicate the ancestral haplotype; e.g. the blue segment in d.alb03 could have resulted from recombination in the either the d.alb03 lineage or lineage ancestral to the rest of the Y1s. The breakpoints of haplotypes are demarcated by dotted lines and the haplotype lengths are drawn below. C. Based on the node of the species split (*s*), nodes of neo-X and neo-Y haplotype splits (*h's*), and neo-X and neo-Y tips, branch lengths are used to approximate the age of the four haplotypes at each window. Left panel, for branch lengths between *s* and *h's*, shorter lengths indicate older haplotypes. Middle and righ panels, for branch lengths between *h's* and their respective neo-X (middle) and neo-Y (right) tips, longer lengths indicate older haplotypes. Schematics of the trees are depicted below each panel, with the branch length of interest marked by dotted lines; n, X and Y, represents *D. nasuta* Chr.3, neo-X, and neo-Y, respectively; 03 represents the Y1-03 haplotype. The number of windows used for each haplotype depends on the length of the haplotype as depicted in 3B. For branch length estimates of the same windows across the haplotypes, see S5 Fig.

different haplotype sizes used for each Y type, we restricted our estimates to windows between 16.2 Mb and 33.2 Mb, where all Y types have a different haplotype; we find the same pattern with Y3 being the oldest and Y2 the youngest (S5 Fig). Based on the ratio of the two branch length measures and the estimated species age of ~250 Kya, we estimate that the Y1 haplotype stopped recombining roughly 107 Kya, and Y3 and Y2 are roughly 28 and 17 Kya older and younger, respectively (Table 1). The different ages of the haplotypes and time of cessation of male recombination may indicate the sequential introduction of the causal allele to suppress male recombination in the different populations.

## Residual *D. nasuta* haplotype on the neo-Y is broken down by male recombination

To better understand the evolutionary history of the neo-sex chromosomes after their formation from an autosome, we compared them to *D. nasuta*'s Chr. 3. We included genome-wide

**Table 1. Age of neo-Y haplotypes.**

| | Median branch length | | Normalized *h*-to-tip[*] | Age (Kya)[**] ± 95% CI |
|---|---|---|---|---|
| | *s*-to-*h* | *h*-to-tip (neo-X) | | |
| Y1 | 0.000490 | 0.000402 | 0.4272 | 106.8 ± 40.2 |
| Y1-03 | 0.000445 | 0.000428 | 0.5064 | 122.1 ± 55.0 |
| Y2 | 0.000586 | 0.000330 | 0.3584 | 89.6 ± 33.8 |
| Y3 | 0.000338 | 0.000446 | 0.5409 | 135.2 ± 54.2 |

[*]h-to-tip length proportional to s-to-tip length

[**]Based on species split of 250kya

polymorphism data for nine *D. nasuta* strains, and inferred 742,084 variant sites between the neo-sex chromosomes and Chr. 3. Based on the prevalent model whereby the neo-X and neo-Y fusions occurred after the species split [33–38], the two neo-sex chromosomes are expected to be more similar to each other than to Chr. 3. The neo-Y shows higher levels of $F_{ST}$ and net nucleotide divergence ($D_A$) from Chr. 3 than the neo-X does for nearly the entire length of the chromosome, possibly due to its slightly lowered π. For most windows along the neo-sex chromosomes (from roughly 15-51Mb), the neo-X and neo-Y are more similar to each other than either is to Chr. 3 of *D. nasuta* (Fig 4A–4C). Both $F_{ST}$ as well as absolute ($D_{XY}$) and net ($D_A$) nucleotide divergence are lower between the neo-X and neo-Y than neo-X/Chr. 3 or neo-Y/Chr. 3, but this difference is much less pronounced from 0-15Mb (Fig 4A–4C), likely due to introgressions or incomplete lineage sorting. Overall, these patterns are consistent with the canonical model of the sex chromosome formation occurring shortly after the species split. Interestingly, however, at the centromere proximal region (51-55Mb), we see a surprising reversal where the neo-X is more differentiated from Chr. 3 (Fig 4A, S6 Fig); this region, as mentioned earlier, also shows sharply elevated $F_{ST}$ between the neo-X and all neo-Y groups (Figs 2C and 4A). Unlike the rest of the chromosome, the net nucleotide divergence ($D_A$) between neo-X and Chr. 3 is also significantly higher at this region than the $D_A$ between neo-Y and Chr. 3 (Fig 4B and 4C, S6 Fig). Compared to the majority of the chromosome arm (15-51Mb), the $D_A$ between neo-Y and Chr. 3 at the centromere proximal region is significantly reduced while the $D_A$ between neo-X and Chr. 3 remains similar across the region (Fig 4B and 4C). These results indicate that the centromere proximal region of the neo-Y is more similar to Chr. 3 than to the neo-X.

We constructed maximum likelihood trees with the neo-Xs, neo-Ys, and Chr. 3s along the chromosome in 200kb sliding windows to decipher the unexpected similarity between the neo-Y and Chr. 3. While the neo-Ys are reciprocally monophyletic to the neo-Xs from 53Mb to the end of the chromosome (Fig 2D), we find two different topologies when incorporating Chr. 3s. For ~0.6 Mb directly next to the centromere, the monophyletic neo-Y clade is sister to the neo-X (Fig 4D and 4H). The topology then changes such that the neo-Y clade clusters with/within the *D. nasuta* clade (Fig 4D and 4G), indicating that the centromere proximal region of the neo-Y contains a recombinant haplotype block that is found in *D. nasuta* (henceforth, *D. nasuta* haplotype), consistent with the $D_A$ and $D_{XY}$ estimates. The *D. nasuta* haplotype is found within all three Y groups and with the same centromere proximal end, and is roughly 4.2 Mb (51.0–55.2Mb) in Y1 and Y2 but 1.8 Mb shorter on Y3 (52.8–55.2Mb). The shorter *D. nasuta* haplotype block on Y3 has a distal end that lands precisely at a haplotype breakpoint (Fig 3B and Fig 4D), arguing that male recombination truncated the segment with sequences from the neo-X. Given that the *D. nasuta* haplotype is found across all Y types, we suspect that the neo-Y fusion happened between the ancestral Y and a recombinant Chr. 3 that already harbored the *D. nasuta* haplotype, instead of recombination after the fusion that introduced a *D. nasuta* haplotype. The recombinant chromosome containing the *D. nasuta* haplotype could have been segregating in the population prior to the species split (incomplete lineage sorting), or produced by hybridization after the split. While no hybrids have been reported or collected in the wild, we find that the neo-X of one of our *D. albomicans* lines (d. alb.SHL2) contains a large introgression from *D. nasuta* (Fig 4D and S7 Fig). This Indian line was collected in the western edge of the *D. albomicans* range overlapping with the *D. nasuta* range, suggesting that hybridization may play a role in the process of speciation.

## No evidence of chromosome-wide down-regulation of the neo-Y

In the absence of male recombination, the neo-Y is expected to start accumulating deleterious mutations. Indeed, genes on older neo-Y chromosomes have been found to be down-regulated

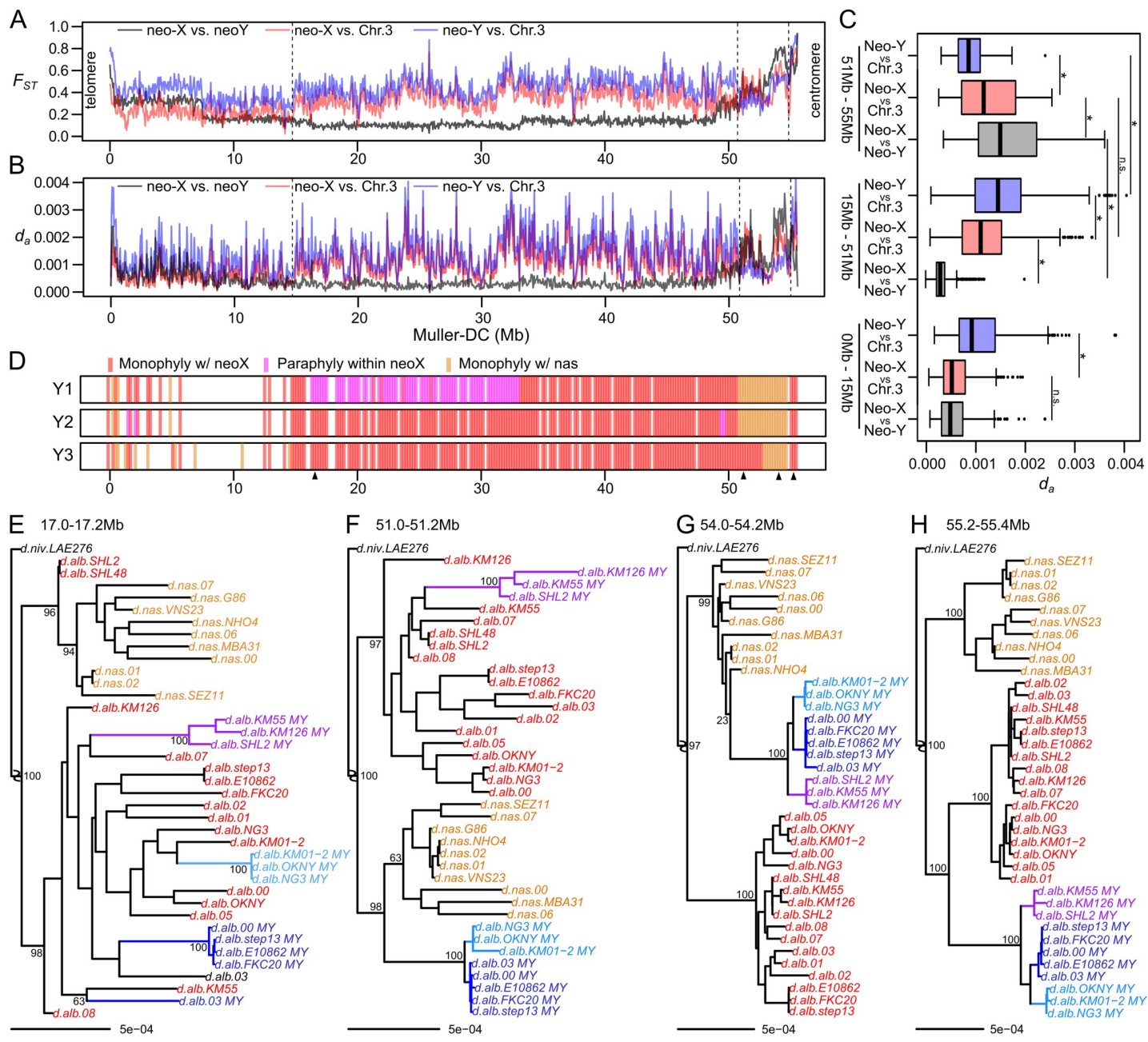

**Fig 4. *D. nasuta* haplotype on the neo-Y.** A. The $F_{ST}$ between the neo-sex chromosomes (gray), the neo-X and *D. nasuta* Chr. 3 (red), and the neo-Y and Chr. 3 (blue) are plotted in the top panel. B. The net divergence ($D_a$) between the neo-sex chromosomes (gray), and of neo-X (red) and neo-Y (blue) versus Chr. 3 are in the lower panel. C. The distribution of $D_a$ (from 4B) from 0-15Mb, 15-51Mb and 51-55Mb are shown in boxplots; * represents p-values < 0.005 (Wilcoxon's rank sum test). D. Based on sliding-window 200kb maximum likelihood trees of neo-Xs, neo-Ys, and Chr. 3s, the phylogenetic relationships between each Y1 group to the neo-X and Chr. 3s are determined. Windows in which the neo-Ys fall within or are sisters to neo-Xs either monophyletically or paraphyletically are plotted in red or magenta, respectively. Windows in which they fall within or are sisters to Chr. 3 are marked yellow. Missing windows are those in which the neo-Xs and Chr. 3s do not sort cleanly into separate clades, likely due to introgression or incomplete lineage sorting. The SHL2 and SHL48 lines are removed from the phylogenies prior to topology inference. E-H Representative windows (demarcated by arrowheads in 4D) of notable topologies are displayed, with the neo-Xs and Chr. 3s colored in red and yellow, respectively; neo-Y individuals are color-coded by their Y group, as in Fig 1. Bootstrap support values are displayed for the species split nodes, the neo-X and neo-Y split nodes, and the neo-Y and Chr. 3 split nodes.

due to pseudogenization and spreading of heterochromatin from adjacent repeats [13,43,57]. The neo-X, in turn, will become dosage compensated through transcriptional up-regulation in males, in order to balance transcriptional silencing of neo-Y homologs [58,59]. A previous study found that the neo-sex chromosomes in *D. albomicans* show widespread down-regulation of neo-Y linked genes, despite containing few pseudogenes. Specifically, 30.1% of genes were found to show significant neo-X biased expression, with the neo-X alleles on average being ~1.3-fold more highly expressed than their neo-Y counterparts [54]. However, our reanalysis of the previously generated RNA-seq data of the KM55 strain, belonging to Y3, reveals little difference between the expression of neo-X and neo-Y alleles, with a median fold-difference of 1.04, a negligible bias that is also observed in the DNA sequence (Fig 5A, Table 2, S8 Fig). Out of 3181 genes with distinguishable neo-sex chromosome alleles, only 115 (3.6%) show significant neo-X bias. Similarly, RNA-seq read counts at individual SNP sites differentiating the neo-X and neo-Y show a median fold-difference of 1, albeit with more noise (S9 Fig). To ensure that our method is able to sensitively detect allele-specific expression, we simulated reads corresponding to 1.2-, 1.25-, and 1.5-fold neo-X bias; we were able to closely recapitulate the expression differences confirming the efficacy of our pipeline (S10 Fig). Consistently, we see no evidence for chromosome wide reduction in expression for Y1 (d.alb.03) and Y2 (d.alb. OKNY-2) (Fig 5A, Table 2), with very few significantly neo-X biased genes and negligible median expression differences between the alleles.

To determine the source of this discrepancy, we examined the allele-specific read counts previously generated [41]. We found a high degree of neo-X bias even among mapping from male DNA-seq (median fold difference of 1.56, S11A Fig), where no bias is expected. This bias is further exaggerated in the male RNA-seq counts (median fold difference of 2.573, S11B Fig). The reference for mapping used previously was from a female assembly that lacked the neo-Y, suggesting that the skewed neo-X counts are the result of reference allele bias. The previous study attempted to correct for this bias by weighting the neo-X and neo-Y RNA-seq counts at each gene by their DNA fold-differences [41]. Despite seeming reasonable prima facie, this method is ineffective in mitigating the reference allele bias (S11C–S11E Fig). Our current approach of generating strain and neo-sex specific references (see Materials and Methods) substantially alleviates reference allele bias in both the DNA-seq and RNA-seq male samples, resulting in low fold-differences of 1.041 and 1.023, respectively (S12 Fig).

In addition to gene expression differences, we compared heterochromatin and dosage compensation profiles of the sex chromosomes. We assayed two histone marks in male *D. albomicans* larvae from the Taiwanese Y1 strain (d.alb.03) using allele-specific ChIP-seq: H3K9me3, which is associated with silencing heterochromatin [57,60,61], and H4K16ac, which is found at the dosage-compensated, hypertranscribed X of male Drosophila [62]. As expected, the repressive histone modification H3K9me3 is enriched at the dot chromosome and the ends of chromosomes corresponding to pericentromeric regions (Fig 5B). The neo-sex chromosomes show no elevation in H3K9me3 enrichment relative to other chromosomes (Fig 5B), and the neo-Y and neo-X show nearly indistinguishable levels of the repressive H3K9me3 mark across the chromosome (Fig 5C), and enrichment levels across chromosome windows are highly correlated (S13 Fig). Thus, global H3K9me3 levels reveal no evidence of neo-Y heterochromatinization. Similarly, we see no bias for the dosage compensation mark H4K16ac along the neo-X chromosome (Fig 5C, S13 Fig), despite substantial enrichment on the old X chromosome (Muller A; Fig 5B). This indicates that the neo-X has not yet evolved dosage compensation through H4K16ac, and is consistent with a lack of systematic up-regulation of the neo-X at the transcript level (Fig 5A). These results show that the neo-sex chromosomes have no clear signs of epigenetic differentiation, consistent with overall similar expression levels of the neo-X and neo-Y.

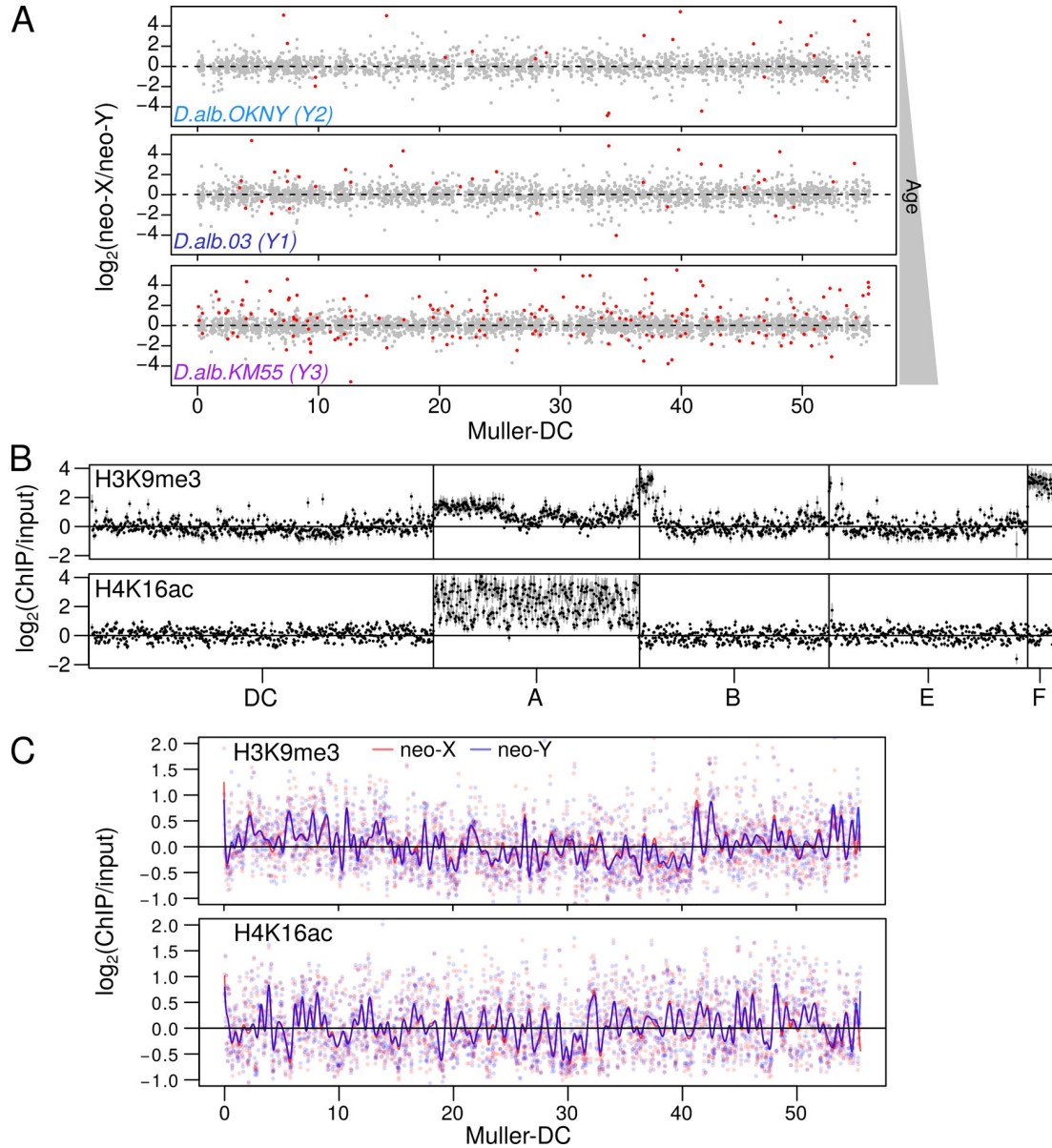

**Fig 5. Absence of chromosome-wide differentiation.** A. Based on allele-specific RNA-seq, the corrected fold-difference (see Materials and methods) between the expression of neo-X and neo-Y alleles of each gene is plotted for neo-Ys of different ages: OKNY (Y2), 03 (Y1), and KM55 (Y3). Red points represent genes with significant allele bias. B. Enrichment of the repressive heterochromatin (H3K9me3, top) and dosage compensation (H4K16ac, bottom) marks for strain 03 are plotted in 50kb windows genome-wide; error bars represent the standard errors inferred from quadruplicates. C. The indistinguishable enrichment profiles of the neo-X (red) and neo-Y (blue) for the two chromatin marks are plotted with Loess smoothing curves overlaid. Correlations of the neo-X and neo-Y enrichments at individual windows can be found in S13 Fig.

## Beginning neo-Y degeneration correlates with neo-Y haplotype age

Theory predicts that non-recombining neo-Y chromosomes should accumulate deleterious mutations, and the amount of degeneration should correspond to its age [42]. As mentioned above, we find that the different neo-Y haplotypes were formed at different time points, and stopped recombining between roughly 90,000 to 135,000 years ago. Despite the absence of chromosome-wide epigenetic differentiation between the neo-sex chromosomes, the number

**Table 2. Differential expression between neo-X and neo-Y alleles.**

| | Median log2(neo-X /neo-Y) | | No. of genes | Significantly differentially expressed alleles** | | |
|---|---|---|---|---|---|---|
| | RNA-seq | DNA-seq | | Both | neo-X biased | neo-Y biased |
| | | | passing filter* | (% of genes) | (% of both) | (% of both) |
| d.alb OKNY-2 (Y2; youngest) | 0.0000 | 0.0123 | 2128 | 26 (1.2%) | 18 (69.2%) | 8 (30.8%) |
| d.alb 03 (Y1; medium) | 0.0498 | 0.0313 | 2391 | 37 (1.6%) | 28 (75.7%)† | 9 (24.3%) |
| d.alb KM55 (Y3; oldest) | 0.0580 | 0.0590 | 3181 | 168 (5.3%) | 115 (68.5%)† | 53 (31.5%) |

* Genes with <5 reads in both alleles are removed

** False discovery rate corrected p < 0.05 (Fisher's Exact Test)

† Significantly more neo-X biased genes p < 0.01 (Binomial Exact Test)

of genes with significant neo-X biased expression is more numerous than neo-Y biased genes for all three neo-Y types (Table 2, Fig 6A; note that this difference is only significant for Y3 and Y1 as they have more genes, p < 0.01, Binomial Exact Test). Most interestingly, the number of neo-X biased genes is correlated with the age of the neo-Y (Table 2, Fig 5A, Fig 6A), suggesting that older neo-Y chromosomes have more genes with reduced expression relative to the neo-X. Thus, neo-X biased gene expression may reflect the beginning stages of differentiation when degeneration of the neo-Y happens on a gene-by-gene basis.

To determine whether older neo-Y types are associated with more deleterious DNA mutations, we focused on the region (7.6–48.6Mb) where all haplotypes differ, excluding the small haplotype found in the Taiwanese line d.alb.03 (Fig 3A). We find that the number of fixed SNPs increases with the age of the haplotypes (Table 3, Fig 6B). While there are few nonsense fixations which are diagnostic of functional degeneration, the number of non-synonymous fixations correlates with age, with Y3 having the most (n = 969) and Y2 having the least (n = 478) amino-acid changes. Likewise, the median rates of synonymous ($K_s$) and non-synonymous substitutions ($K_a$) per gene, and their ratio ($K_a/K_s$), increase with age of neo-Y types (Fig 6D–6F). This suggests that more mildly deleterious amino-acid mutations have accumulated on older Y types. Given few nonsense mutations, we additionally used the number of neo-Y specific indels as a proxy for functional decay of the neo-Y. Indeed, the oldest Y3 haplotype has the most fixed indels (n = 11,108), the intermediate Y1 has 5,615, and the youngest Y2 has the fewest indels (n = 3,385), but only a small fraction of indels are found within coding sequences (Table 3). Similarly, the oldest haplotype Y3 has the most number of indels that are within coding sequences (n = 163) and that create frameshifts (n = 49) while the youngest haplotype Y2 has the fewest (Table 3, Fig 6C). We see the same trend with frameshift within 500bp of genes, some of which are likely to cause regulatory changes. Despite the small number of indels that are likely to be disruptive, these results demonstrate that the extent of degeneration in these young neo-Ys is correlated with their age. Curiously, haplotype-specific indels and SNPs appear to be unevenly accumulated across the chromosome (S14 Fig). Despite having fewer indels for most of the chromosome, Y1 appears to have similar numbers to that of Y3 up to ~15Mb. Additionally, Y2 has a spike of indels between 27–29 Mb which may represent a larger structural variant.

## Independent and gene-by-gene downregulation of neo-Y alleles

Interestingly, no neo-X biased genes are shared across all three neo-Y groups and no more than four are shared between any two neo-Y groups (Fig 7A), arguing that they are independently degenerating. The lack of shared loci of neo-sex chromosome differentiation suggests that degeneration began only after the establishment of the locked neo-Y haplotypes when

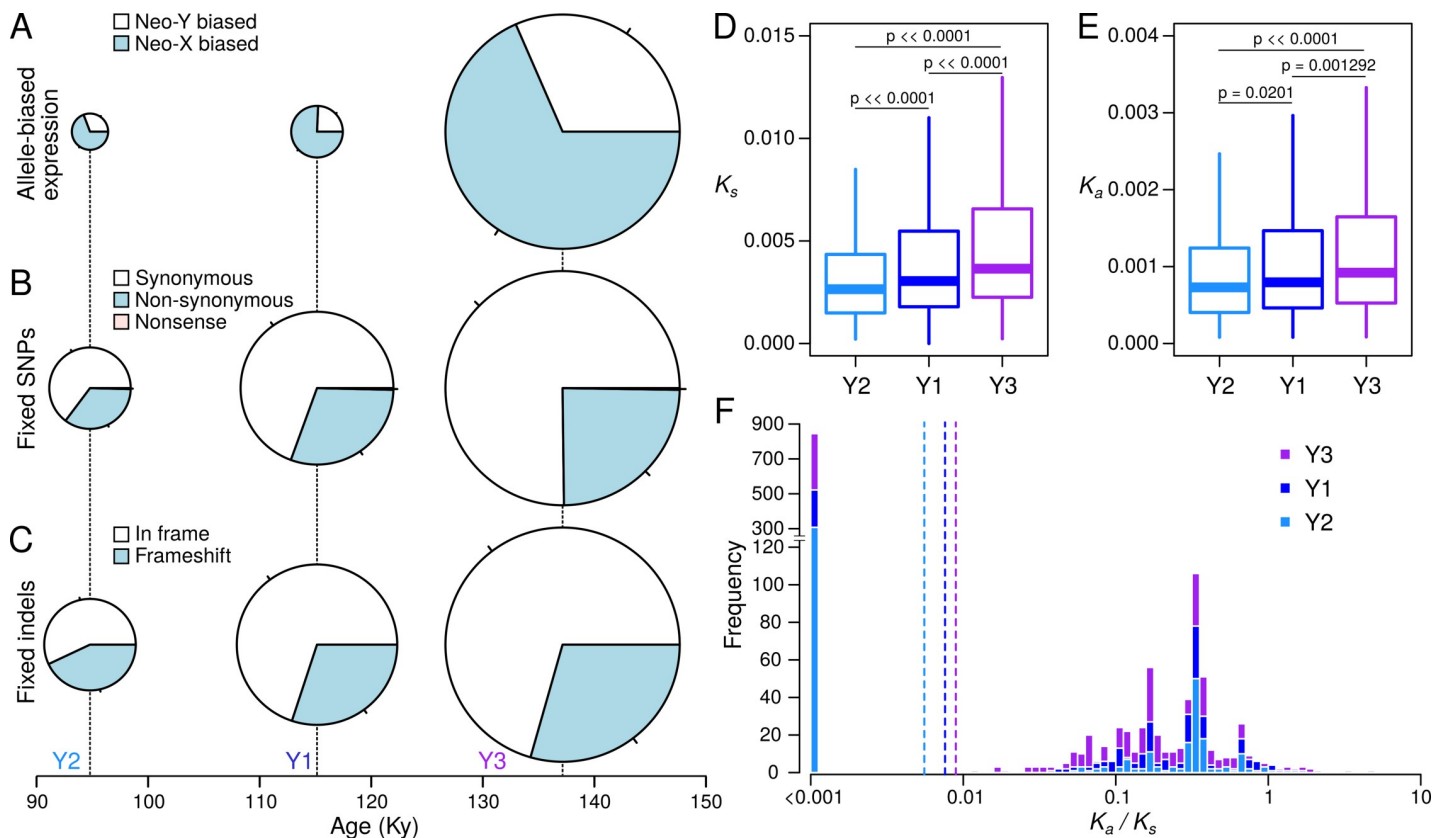

**Fig 6. Extent of neo-Y degeneration is correlated with haplotype age.** A. The proportion and number of genes with significant neo-X and neo-Y bias are plotted for each of the three haplotypes. B. Same as A, but with different classes of fixed and haplotype-specific SNPs found in CDS. C. Same as B, but with indels found in CDS. Size of pie charts are proportional to the total number of genes/mutations for each haplotype. D-E. Distribution of the rates of synonymous ($K_s$) and non-synonymous ($K_a$) fixed substitutions among the Y-types. F. Distribution of $K_a/K_s$ ratios for each of the three Y-types in log scale. Dotted vertical lines demarcate the average ratio.

male recombination ceased. Furthermore, we find that across all neo-Ys, the neo-X biased genes are significantly depleted of testes-biased genes and enriched for somatic genes (Fig 7B). To ensure that the paucity of testes-biased genes is not a technical artifact of using RNA-seq from head samples, we identified neo-X-biased genes in the testes of KM55 and similarly found significant depletion. Despite independent degeneration of the neo-Y types, genes with male-specific function are thus likely to be shielded from the initial accumulation of deleterious alleles.

To determine whether the neo-X bias is due to up-regulation of the neo-X alleles or down-regulation of the neo-Y alleles, we conducted reciprocal crosses producing F1 hybrids between *D. albomicans* and *D. nasuta*; sons sired by *D. albomicans* have neo-Y and Chr. 3 while sons sired by *D. nasuta* have neo-X and Chr. 3, thus allowing us to determine expression differences

**Table 3. Degeneration of neo-Y haplotypes.**

| | No. of substitutions (7.6–48.6Mb) | | | | | No. of indels (7.6–48.6Mb) | | |
|---|---|---|---|---|---|---|---|---|
| | **All** | **Within CDS** | **Syn.** | **Non-syn.** | **Nonsense** | **All** | **Within CDS** | **Frame shifts** |
| Y1 | 13274 | 2572 | 1786 | 777 | 9 | 5615 | 163 | 49 |
| Y2 | 7480 | 1370 | 887 | 478 | 5 | 3385 | 93 | 40 |
| Y3 | 20263 | 3942 | 2694 | 969 | 9 | 11108 | 238 | 70 |

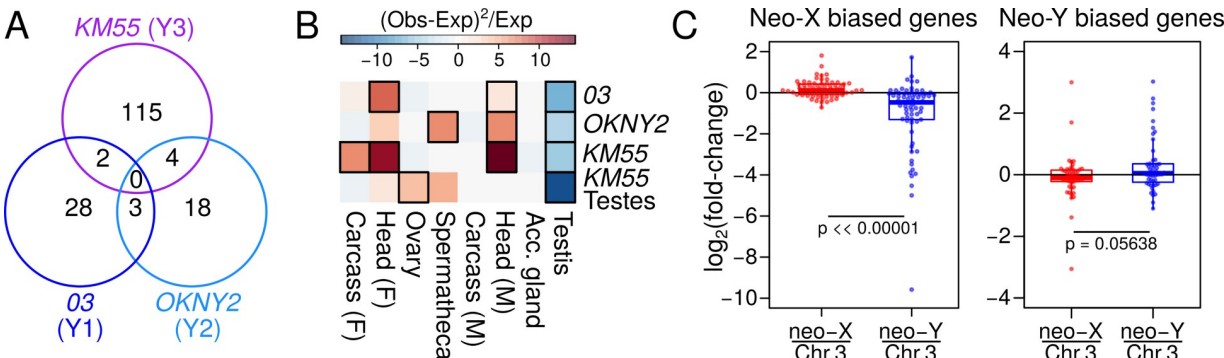

**Fig 7. Independent gene-by-gene downregulation of neo-Y alleles.** A. Venn diagram illustrates the number of neo-X biased genes that overlap between the neo-Y haplotypes. B. The heatmap depicts over- and under-representation of tissue-bias among neo-X-biased genes among the different neo-Ys. Colors represents the Chi-square statistics, modified to include enrichment (+) or depletion (−); significant cells are boxed with black borders (p < 0.05, see materials and methods). To expand the number of genes for this analysis, we used a less stringent cutoff (p-value < 0.1) than the one used in Table 2. C. For genes with neo-sex bias in strain *D.alb.03*, we determined their allele-specific expression in reciprocal hybrid males with Chr. 3 and either neo-X or neo-Y. The distribution (boxplot) and the gene-by-gene (points) fold-difference of the alleles are plotted in log scale for the neo-X biased genes (left) and neo-Y biased genes (right). For the neo-X biased genes, neo-X and Chr. 3 alleles have similar expression level, but the neo-Y allele is on average lower than the Chr. 3 allele. For the neo-Y biased genes, both the neo-X and neo-Y alleles are similar to the Chr. 3 alleles. P-values are based on Wilcoxon rank sum test.

that are due to cis-regulatory divergence between the neo-X, neo-Y, and the ancestral Chr. 3 [63]. Down-regulated neo-Y alleles will have lower expression than the neo-X and, importantly, Chr. 3 alleles. To increase the power of this analysis, we expanded the number of sex-biased genes by using a less stringent p-value cutoff of < 0.1, instead of the < 0.05 previously used (Table 2); this increased the number of neo-X biased and neo-Y biased genes to 69 and 35, respectively. For the genes with neo-X bias, the neo-X and Chr. 3 alleles have similar expression levels, but the expression of neo-Y alleles are on average half (mean = 0.48) of their Chr.3 counterparts (Fig 7C, left). These results reveal that neo-Y down-regulation, not neo-X up-regulation, primarily accounts for the neo-X bias, consistent with gene-by-gene degeneration of the neo-Y alleles.

Roughly 30% of genes with differential expression between the neo-sex chromosomes show neo-Y biased expression (Table 2). Neo-Y biased expression of genes could reflect male-specific selection on the male-limited neo-Y (that is, the gene content on the neo-Y may become more male-biased or 'masculinized'), or reduced selection to maintain expression of the neo-X. We find that neo-Y biased genes have not changed their expression on the neo-Y, relative to Chr. 3, but instead appear to have slightly (but not significantly) reduced expression on the neo-X (Fig 7C). Thus, while the overall number of neo-Y biased genes is small (Table 2), comparison to *D. nasuta* suggests that neo-Y bias is not due to masculinization of the neo-Y, but reduced expression from the neo-X.

## Discussion

### Multi-step model for generating multiple degenerating neo-Y types with a single fusion

Neo-Y chromosomes of Drosophila have served as prominent models to study the evolutionary and molecular processes resulting in the degeneration of a non-recombining chromosome [4]. Unique to *D. albomicans*, the origination of its neo-Y chromosome occurred in an ancestor without male achiasmy, and our population genetic and phylogenetic analyses shed light on the complex evolutionary history of *D. albomicans*' neo-Y chromosome. We found that the

presence of male recombination created different neo-Y haplotypes that are now geographi-cally distributed in different parts of the species range, and propose a multi-step model to account for our results and previous models of the karyotypic evolution of *D. albomicans* (Fig 8). Laboratory crosses between *D. albomicans* and *D. nasuta* have suggested that the fusion between Chr. 3 and the X is more likely to be established first, as opposed to a Chr. 3 –Y fusion [53]. After the Robertsonian fusion, the neo-X can recombine with Chr. 3 in a meiotic trivalent both in females and males prior to the reversal to achiasmy (Fig 8A). The second fusion, form-ing the neo-Y, occurred between the Y and a Chr. 3 that harbored a large *D. nasuta* haplotype near the centromere. This fusion is likely under positive selection to resolve mis-segregation due to the trivalent pairing during meiosis [53]. While the fusion produced a severe bottleneck, the presence of male recombination allowed the neo-Y to regain diversity through exchange with the neo-X, which also broke down the *D. nasuta* haplotype. However, once male recombi-nation stopped in a particular population, the diversity on the neo-Y rapidly decreases, eventu-ally leading to the fixation of one particular Y type in the population, due to drift and possibly positive selection for male beneficial alleles (Fig 8B). Unlike on recombining chromosomes, deleterious mutations cannot be unlinked on the neo-Y and will increase in number and fre-quency in the population (see introduction). Because different isolated populations would likely fix different neo-Y haplotypes, overall diversity can be maintained on the neo-Y, despite high differentiation among haplotypes (Fig 2B). Our dating of neo-Y haplotypes suggests that male recombination stopped at different times for the different Y haplotypes. This raises the possibility that the mutation causing the cessation of male recombination in *D. albomicans* spread sequentially from mainland Asia through the islands.

The remnant of the *D. nasuta* haplotype near the centromere raises the question of why the *D. nasuta* segment was not completely lost while the neo-Y was recombining. Recombination rate around the region was likely low due to the proximity to the pericentromere where cross-overs are suppressed by the effects of pericentromeric heterochromatin and/or centromere. Consistent with the suppression of recombination near the pericentromere, no recombinants were recovered within this region in a cross scheme to generate introgressions between the two species [64]. However, given that the Y3 population has a shorter *D. nasuta* haplotype resulting from recombination with the neo-X at the distal end of the haplotype, recombination within the region is likely possible but rare (Fig 4C). Furthermore, the *D. nasuta* haplotype does not extend into the centromere, but instead is gapped by a short stretch of neo-Y sequence that is more closely related to the neo-X; this entails that a crossover close to the peri-centromere introduced the *D. nasuta* haplotype prior to the neo-Y fusion (Fig 4C). If this region can readily recombine, the fixation and maintenance of the haplotype may instead sug-gest the interesting possibility that a male-beneficial alleles resides within the region and that the *D. nasuta* haplotype represent an adaptive introgression [65,66].

## Cessation of male recombination initiates degeneration

To date, the neo-X and neo-Y of *D. albomicans* are among the youngest sex chromosomes to be investigated in detail. Overall, we find no global differences in levels of expression between neo-sex linked genes, and nearly indistinguishable heterochromatin and dosage compensation profiles between the neo-X and neo-Y. However, expression differences at individual genes and biased accumulation of amino-acids and indels suggests that the neo-Y chromosomes are showing early signs of degeneration, and the amount of differentiation is correlated with the Y ages. In particular, we identified four neo-Y haplotypes of different age that are geographically distributed. The oldest Y3 haplotype has the highest number of fixed nonsynonymous changes, indels, and neo-X biased genes, indicating the most extensive degeneration; the younger Y1

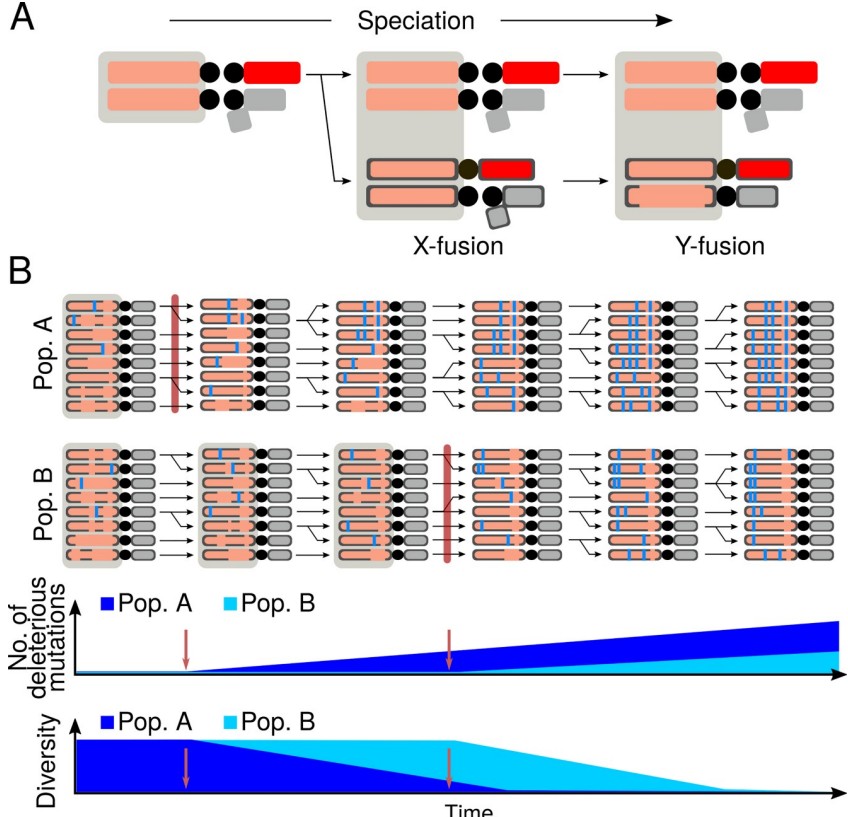

**Fig 8. Multi-step model for neo-Y evolution in *D. albomicans*.** A. The sequential fusions of the neo-sex chromosomes during speciation are depicted in the schematic. *D. albomicans* chromosomes have black outlines, to differentiate from *D. nasuta* chromosomes. Chromosomes that can recombine with each other are encompassed by gray boxes. Following the neo-X fusion, the neo-Y emerged via fusion with a chromosome that harbors a block of *D. nasuta* haplotype (no black outlines). B. Prior to the cessation of male recombination, the diversity of neo-Y is primarily the product of recombination with the neo-X and Chr. 3. During this time, the *D. nasuta* haplotype on the neo-Y is broken down by recombination with the neo-X. After males stop recombining in the population (maroon bar), the diversity is rapidly lost leading to the fixation of different haplotypes in different populations and deleterious mutations (blue bars) begin to accumulate on the neo-Y. The population in which male recombination stopped earlier (Pop. A) is expected to have more degenerate neo-Ys as compared to the population in which recombination stopped later (Pop. B). Bottom graphs track the accumulation of deleterious mutations and reduction in diversity in the two populations over time. Cessation of male recombination in the two populations are marked by red arrows.

and Y2 groups show less extensive degeneration. The paucity of overlapping genes that are neo-X biased between the haplotypes indicates that degeneration began only after the haplotype became locked when males stopped recombining, as expected. We show that neo-X biased expression is primarily due to down-regulation of the neo-Y alleles, as revealed by comparison with *D. nasuta*'s Chr. 3 in hybrids. At this early stage of sex chromosome differentiation, down regulation of neo-Y genes is likely to be on a gene-by-gene basis. Indeed, we find several genes with significantly higher levels of neo-X expression that also have fixed indels that cause frameshifts on the neo-Y in all three Y groups (Table 2). Unlike the typical trajectory of a neo-Y in Drosophila where male-exclusive transmission results in the immediate cessation of recombination, meiotic exchange prior to the halt of male recombination maintained genetic diversity on the neo-Y chromosome and likely prevented the accumulation of deleterious alleles. The age of a neo-Y haplotype appears younger and less degenerate than expected based on when the neo-Y fusion occurred and instead depends on when male recombination was abolished.

While differentiation of the neo-Y is atypical compared to other Drosophila, it is similar to Y formation in organisms with male recombination in both animals and plants [7,8,10,11,24–28]. The neo-Y chromosome of the Japan Sea Stickleback (*Gasterosteus nipponicus*) formed via a Y-autosome fusion less than 2 million years ago and shows little degeneration outside of the non-recombining region; inside the non-recombining region near the fusion, 15% of the mutations on the neo-Y are deleterious [67]. Similarly, the pseudoautosomal region on the mammalian Y (PAR-Y) chromosome retains the ability to pair and recombine with the X counterpart, despite having diverged over 180 million years ago [68–70]. Although the bulk of the chromosome is highly degenerate, recombination allows the PAR-Y to maintain genes important for development and cellular function [71]. Outside of the PAR-Y recombination with the X became suppressed due to multiple large-scale inversions [72,73]. Our study on the formation of the neo-Y in *D. albomicans* offers a unique look at the transition of recombination suppression and provides insight on the rate of degeneration immediately after this transition.

## Geographically restricted distribution of neo-Y haplotypes that are independently degenerating

After the cessation of recombination, diversity is expected to rapidly diminish due to a combination of Hill-Robertson effect, Muller's ratchet, and selective sweeps. Indeed, our results revealed that individuals within a geographically isolated region have nearly identical neo-Ys belonging to the same haplotype. While more sampling of neo-Ys is required to fully capture the geographic distribution of different Y types, our results paint an intriguing picture of independent fixations of different haplotypes in different isolated populations followed by independent initiation of degeneration. Given that the different haplotypes differ in age, the cessation of recombination likely spread from the oldest population in Indochina (Y3), to Taiwan (Y1), to Okinawa (Y2). The mainland Japan population appears to be a recent invasion from the Taiwanese population [47], supporting the similarity between the Y1 and Taiwanese haplotypes. Under this model, the sequential spread of the allele(s) to suppress male recombination is likely aided by positive selection, as the fixation has to take place in each population to result in the species-wide fixation as currently observed [56].

Our results also show that the neo-Y groups share very few indels and genes that are downregulated on the neo-Y, indicative of independent degeneration. However, the small sets of genes undergoing degeneration, remarkably, show a similar attribute: they are all depleted for genes with testes-biased expression. This is consistent with findings that male-biased genes are maintained for longer periods of time on the degenerating neo-Y-chromosome of *D. miranda*, likely due to selection to preserve genes important for male-exclusive function [74]. Thus, while the independent degeneration of multiple neo-Y haplotypes may be unique given the unusual evolutionary history of *D. albomicans*, the pattern of degeneration follows a typical and generalizable trajectory.

## Materials and methods

### Genotyping samples from whole genome sequences

See S1 Table for all strain and sequence information. For whole genome sequencing, 5 sexed flies were collected and their DNA extracted using the DNeasy Kit (Qiagen). Libraries were generated using the Illumina TruSeq Nano DNA kit with 550bp size selection and sequenced on the HiSeq 4000 High Output mode by the Vincent J. Coates Genomics Sequencing Laboratory at the University of California, Berkeley. All samples were mapped with bwa (v.0.7.15)

mem [75] to the reference generated by [76] under default settings for pair-end reads. We then processed the mapped reads according to GATK best practices which includes removing duplicates with MarkDuplicates in Picard tools (v.2.18.14) and sorting and merging with Samtools (v.1.5) [77,78]. We genotyped each chromosome arm separately using GATK's (v3.8.0) HaplotypeCaller under default parameters. Note, because of a known but unaddressed bug in GATK where it would randomly crash, we ran HaplotypeCaller in 5 Mb windows (using the -L setting) for each chromosome to avoid restarting the entire run every time the program crashes. The genotype files (.vcf) are then merged together after all calls are complete.

Depending on the samples required for the analyses, we first filtered out the samples that are unnecessary, followed by removal of the indel sites and sites with missing genotypes. We then filtered for a minimum coverage of 5x and genotype quality of 20. For example, for the phylogenies with only the neo-X and neo-Y, we kept only *D. albomicans* samples and the one outgroup before subsequent filtering. This is to ensure that we are maximizing the number of sites we keep for each analysis, since the probability of one sample failing to pass the filters, which will cause the site to be removed, increases with more samples. All filtering was accomplished using bcftools (v.2.26.0) [79].

## Inferring and validating the neo-Y genotypes

For each strain, we generated a vcf file containing both the female and male genotypes from which we identified sites that are homozygous in the female but heterozygous in the male. If one of the heterozygous alleles in the male is the same as the female allele, the neo-Y genotype is then the other allele. Sites that are heterozygous in the females are left as missing in the neo-Y and these sites are subsequently removed.

To determine the efficacy of this computational pipeline, we identified HindIII cut sites on the neo-X of the reference strain by grepping for the restriction sequence AAGCTT then selecting those that are disrupted by neo-Y SNPs. Primers flanking the cut sites were designed using NCBI's Primer-BLAST (S2 Table) [80] and ordered from IDT. PCR amplicons from female and male gDNA were digested with HindIII overnight at 37˚C. Females, homozygous for the cut sites, are expected to have fewer bands than the males which are heterozygous for the cut and uncut alleles (S2 Table).

For Y-linked indels, we used the same strategy, but also included male-specific homozygous indels as Y-linked. When multiple indels are within 50bp of each other, only the first one is kept to avoid mapping errors near indels. Frameshifts are categorized as indels within CDSs that are not multiples of 3.

## Population genetic estimates

Both $F_{ST}$ and $\pi$ were estimated using vcftools (v.0.1.15) with the parameters—window-pi and —fst-window-size in 50kb windows [78]. $D_a$ was calculated from $D_{XY}$ estimated using the popgenwindows.py from the genomics_general package by Simon Martin (https://github.com/simonhmartin/genomics_general), with the formula $Da = D_{XY} − (\pi_X + \pi_Y)/2$ where $X$ and $Y$ are the two populations. Number and proportion of shared polymorphisms between the neo-X and neo-Y were calculated using a custom perl script.

## Generating strain-specific references

We isolated each sample (or neo-X/neo-Y) into its own vcf and removed all homozygous reference sites. Using GATK's FastaAlternateReferenceMaker we generated a "pseudo-reference" fasta allowing for IUPAC ambiguities. The names of the chromosomes are then modified to include the strain information.

## Phylogenetic Analyses

The individual pseudo-references are divided into 200kb windows using samtools faidx and concatenated into one fasta for each window. Maximum likelihood trees were generated using RAxML (v.8.2.11) [81] with the command:

raxmlHPC-PTHREADS-AVX -f a -x 1255 -p 555 -# 100 -m GTRGAMMA -s input.fa -n output.tree -o root_species.

The resulting trees were visualized using the program Densitree [82] or plotted in R with the package Phytools [83]. To evaluate tree topologies, we used the R packages ape and phytools [83,84], in which the is.monophyletic() function was used to determine whether the neo-Ys are monophyletic.

## Assigning haplotypes to individual neo-Ys and estimating their ages

For each 200kb sliding window tree, the number of clades that contain exclusively (monophyletic) neo-Ys are first determined. Neo-Ys that are in the same monophyletic clade are then assigned the same haplotype, and different haplotypes are given a different color in the diagram. To avoid interruption of haplotypes by erroneous trees, any solo window flanked by a different topology on both sides is assigned matching topology to the neighbors. For the age estimates of each haplotype, branch lengths between nodes are measured for each window using phytools. The *s* node (species split node), is determined based on the last common ancestor of the *D. albomicans* and *D. nasuta* individuals using the phytools function mrca(). Windows where the two species fail to sort into distinct monophyletic clades are excluded from the analyses, as *s* cannot be correctly inferred. The *h* nodes (most recent neo-X and neo-Y haplotype splits) are identified as the nodes preceding each of the monophyletic Y group clades using the phytools function getParent(). The neo-X sisters to the Y groups are identified using getSister(). *h*-to-tip lengths are averaged when there are multiple tips (individuals). Estimates reported in Table 1 are the medians of the branch lengths across the appropriate windows of the haplotypes.

## Y-type-specific fixed sites and *Ka/Ks*

Derived neo-Y SNP variants (variants absent in neo-X and Chr. 3) are further divided into variants fixed in each of the three Y types. For Y1, d.alb.03 was removed due to the additional haplotype in the middle of the chromosome (Fig 3). The Y-type-specific fixed variants are then used to generate pseudo-references as above. Coding sequences were extracted based on the gene annotation file [76](.gff) for each of them and the neo-X reference with gffread [85]. For each gene, pairwise alignment files were made between the neo-Y sequences and neo-X sequences. Note, this did not require an aligner, as the neo-X and neo-Y CDS sequences have identical start and stop positions, given that the same gff file was used to extract them. For each Y-type, genes that are identical to the neo-X copies are removed. We then used KaKs_-Calculator [86] to estimate the Ka, Ks, and Ka/Ks, with the flag -m GMYN for the Gamma-MYN algorithm [87].

## Allele-specific RNA-seq and ChIP-seq

RNA were extracted from 10 male heads with Trizol, and libraries were prepared using the TruSeq Stranded Total RNA kit (Illumina). Pair-end reads are first aligned to the reference under default settings with bwa mem. Reads aligning to the neo-X which includes both neo-X and neo-Y reads are then extracted using samtools, and converted back into fastq with the

bamtofastq function in Picard Tools. The fastq reads are then mapped to an index containing both the strain-specific neo-X and neo-Y sequences, and only uniquely mapping reads are kept. Number of reads with allele-specific mapping in each sample is documented in S4 Table. The read counts are then tallied using featureCounts in the Subread (v.1.6.2) package [88], with an annotation file (.gtf) where the chromosome names are modified to match the names of the strain-specific neo-X and neo-Y sequences. Genes with fewer than 5 reads in both alleles are removed from our analysis. The read counts of DNA-seq samples were processed identically. For the corrected fold difference, the fold-difference from the RNA-seq data are divided by the fold-difference from the DNA-seq data. We tested the efficacy of this approach by simulating neo-X and neo-Y reads from D.alb.KM55 at 12:10 (1.2-fold), 10:8 (1.25-fold), and 10:5 (2-fold), coverage ratios, with the package ART [89], respectively, and were able to recapitulate the fold differences at the gene level (S8 Fig). Reciprocal hybrids were generated by mating 5-day old virgin D.alb.03 with male D.nas.00 or vice versa. The RNA-seq data were processed similarly with the addition of D.nas.00 specific reference.

For ChIP-seq, single male larvae were selected and processed following the low-input native chromatin immunoprecipitation protocol [90] with H3K9me3 and H4K16ac antibodies from Diagenode. The libraries were prepared using the SMARTer Universal Low Input DNA-seq Kit (Takara, formerly Rubicon). We mapped the reads similarly to the allele-specific RNA-seq, but used the genomeCoverageBed from bedtools (v.2.26.0) [91] to convert the uniquely mapped neo-X and neo-Y reads into read counts across the chromosome. We then extracted the sites that differ between the neo-X and neo-Ys based on the vcf. For both the input and the immunoprecipitation samples, the read counts at each site were then normalized by the median autosomal coverage. The enrichment at each window is averaged across the sites within the window.

## Tissue enrichment and depletion

Available tissue specific RNA-seq [92] were mapped to the reference genome and the read counts at genes are converted into RPKM. The tissue-specificity, tau, is then determined [93], and genes with tau > 0.5 are deemed tissue biased and assigned to the two tissues with the highest expression. The overlap of tissue-biased and neo-X biased genes are then determined. To increase the power of this assay, instead of using the small number of neo-X-biased genes that pass the cut-off of multiple-testing (FDR) corrected p-value of 0.05, we used those that pass the nominal p-value of 0.01, increasing the number of neo-X-biased genes to 69 (alb.03), 61 (alb.OKNY), and 180 (alb.KM55). For any given Y-type-by-tissue overlap, the expected count is generated from 100,000 rounds of simulations that make two independent random draws without replacement: the number of tissue-biased genes from the set of all genes on Chr. 3 and the number of neo-X-biased genes from the set of genes passing filter (see Table 2, column 4). The number of overlapping genes is determined from the two draws. The expected count is then the median of the distribution from the simulations and the P-value is determined from the percentile of the actual/observed counts.

## Supporting information

**S1 Table. Sample information.**
(XLSX)

**S2 Table. Primers for RFLP sites (HindIII) distinguishing neo-sex chromosome.**
(XLSX)

**S3 Table. Number of male-specific sites across chromosomes.**
(XLSX)

**S4 Table. Neo-sex chromsome specific RNA-seq read counts.**
(XLSX)

**S1 Fig. The neo-X phylogeny of strains with neo-Y genotypes.** Strains are color coded based on their neo-Y type.
(PDF)

**S2 Fig. $F_{ST}$ between different Y populations.**
(PDF)

**S3 Fig. $F_{ST}$ between different Y populations and neo-X.**
(PDF)

**S4 Fig. $F_{ST}$ and $\pi$ at the centromere proximal region.**
(PDF)

**S5 Fig. Same as Fig 3C but estimates are restricted to windows between 16.2 Mb and 33.2 Mb, where all Y types have a different haplotype.**
(PDF)

**S6 Fig. $F_{ST}$ and $D_{XY}$ between the neo-X, neo-Y, and Chr. 3 the centromere proximal region.**
(PDF)

**S7 Fig. *D. nasuta* haplotype on the neo-X of SHL-2.** Windows where SHL-2 falls in the *D. albomcians* neo-X clade are colored red. Windows where SHL-2 falls in the *D. nasuta* Chr.3 clade are colored in yellow.
(PDF)

**S8 Fig. Distribution of allele-specific gene expression across neo-Y chromosomes of different ages.**
(PDF)

**S9 Fig. Allele-specific differential expression at neo-X and neo-Y SNP sites.** Left panels, allele-specific read counts over strain-specific SNP sites (points) differentiating the neo-X and neo-Y chromosomes were used to calculate the fold difference. Right panels, histograms of the distribution of the log2 fold differences. Red lines demarcate the median fold difference.
(PDF)

**S10 Fig. Allele-specific expression on simulated KM55 data.** Neo-X and neo-Y reads were simulated at three different coverage ratios: 10x:5x (2-fold), 10x:8x (1.25-fold), and 12x:10x (1.2- fold). Fold-difference of allele specific read counts at each gene is plotted in log scale. Red dotted lines demarcate the median fold differences, and black dotted lines mark no expression difference. Across multiple levels allele-specific differences, our current pipeline is able to recapitulate the expected ratios, indicating that the lack of neo-X bias is not due to poor sensitivity in our pipeline.
(PDF)

**S11 Fig. Sources of discrepancy with Zhou and Bachtrog 2012.** In Zhou and Bachtrog 2012, while the allele-specific expression for a large number of genes (n = 4839) were determined, only a small subset was used for the analysis (n = 805) after filtering out genes with fold differences (neo-X/neoY) greater than 1.25 or less than 0.75 in the male DNA. The purpose of this filter was to remove genes with substantial allele-bias at the DNA level, where the neo-X and

neo-Y counts are expected to be highly similar. After reanalyzing the read count data generated by Zhou and Bachtrog 2012, the pipeline appears to produce extensive neo-X bias even from the DNA with a median fold difference of 1.56 (A); this is likely the result of reference allele bias, as the reference was generated from females, and therefore only contains the neo-X (also see S12 Fig). The allelic difference is further exaggerated in the RNA with median fold difference of 2.573. The filter therefore, at face value, seems like a sensible strategy to avoid genes with strong technical bias resulting from the pipeline. However, it substantially limited the number of genes being analyzed and reported, with only 16% of the genes being examined. This accounts for the large discrepancy between the number of genes examined between Zhou and Bachtrog 2012 and our study. In addition, Zhou and Bachtrog also attempted to correct for the bias by subtracting out the fold difference in the DNA from that of the RNA, reasoning that the reference allele bias should have similar effect for the DNA and RNA (panel B). Again at face value, this seems like a reasonable approach, but upon revisiting this correction, we do not think it is adequate. First the fold difference at the DNA level is positively but very poorly correlated with that of the RNA ($R^2 = 0.039$, panel D). This argues that the former is a poor predictor of the latter. After the correction, the correlation becomes negative, with a equally poor $R^2$ suggesting that the approach is performing poorly at correcting for the bias (panel E). The distribution of the fold difference at the DNA level is a combination of both the stochasticity in DNA amplification during library prep as well the technical biases introduced by the pipeline. The correction is implicitly assuming that only technical bias is contributing to the variance in the fold difference in the DNA and is to be subtracted from the RNA. This is also apparent when looking at the effect the correction has on the filtered genes where the correction has minimal effects on the fold difference of the filtered list of genes (panel C). In short the pipeline used by Zhou and Bachtrog introduced a substantial amount of reference allele bias that affected both the allele specific read counts in the male DNA and RNA and their approach of correcting for this was insufficient. The use of only one reference for allele-specific expression causes significant reference allele bias (see Stevenson, Coolon & Wittkopp 2013 and also S12 Fig). We therefore generated separate reference sequences for the neo-X and neo-Y. This substantially alleviated the neo-X bias as the median fold differences between the alleles across all male DNA samples are less than 1.05.
(PDF)

**S12 Fig. Reference allele bias when mapping to only the neo-X chromosome.** The high reference allele bias in Zhou & Bachtrog 2012 likely stem from the fact that reads derived from the neo-Y were only mapped to the neo-X, as the neo-Y was not assembled. In our current study, we generated line-specific "psuedoreferences" of both the neo-X and neo-Y, as to minimize reference allele bias. To demonstrate the extent to which reference allele bias affects allele-specific expression in this case, we, using the same DNA-seq and RNA-seq reads as Zhou & Bachtrog, first distinguished the neo-X and neo-Y reads by mapping to the two neo-sex pseudoreferences. We then remapped the neo-Y-mapping reads, to either the neo-X or neo-Y pseudo-references. This remapping was done using Tophat2 (instead of BWA like the rest of our study) to more closely emulate what was done by Zhou & Bachtrog 2012. After mapping to the neo-X reference, we compared the read counts of these neo-Y reads to the neo-X reads to infer allele-specific differences at genes. In both the DNA (panel A, blue) and RNA (panel B, blue) samples, we detected a significant bias for the neo-X allele (median log2 fold-change of 0.292, blue dotted line in Figure A), with the RNA sample showing stronger bias (median log2 fold-change of 0.460, blue dotted line in panel B). In contrast, when the same neo-Y reads were mapped to the neo-Y reference, the allele-specific differences are drastically reduced in both samples (red in panels A and B). Note, we are unable to recapitulate the

severe extent of neo-X bias in both the DNA and RNA reported in Zhou & Bachtrog 2012 despite starting from the same raw reads. Given that an old version of Tophat was used (the no longer supported Tophat1), we suspect that more neo-Y reads failed to map, further exacerbating the reference/neo-X allele bias.
(PDF)

**S13 Fig. Correlations of epigenetic mark enrichment on the neo-X and neo-Y.** Red lines demarcate the identify lines (x = y).
(PDF)

**S14 Fig.** Distribution of fixed derived SNPs (top) and indels (bottom) on the three neo-Y haplotypes: Y1 (blue), Y2 (light blue), Y3 (purple).
(PDF)

## Acknowledgments

We thank Dr. R. Bracewell for helpful discussion on generating phylogenies, D. Mai for discussions pertaining to the *D. albomicans* genome assembly, and A. Nguyen on help with RNA-seq library preparation. We thank the five anonymous reviewers for insightful comments on the manuscript.

## Author Contributions

**Conceptualization:** Kevin H-C. Wei, Doris Bachtrog.

**Data curation:** Kevin H-C. Wei, Doris Bachtrog.

**Formal analysis:** Kevin H-C. Wei.

**Funding acquisition:** Doris Bachtrog.

**Investigation:** Kevin H-C. Wei, Doris Bachtrog.

**Methodology:** Kevin H-C. Wei.

**Project administration:** Doris Bachtrog.

**Resources:** Doris Bachtrog.

**Software:** Kevin H-C. Wei.

**Supervision:** Doris Bachtrog.

**Validation:** Kevin H-C. Wei, Doris Bachtrog.

**Visualization:** Kevin H-C. Wei.

**Writing – original draft:** Kevin H-C. Wei, Doris Bachtrog.

**Writing – review & editing:** Kevin H-C. Wei, Doris Bachtrog.

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
