## [Decision Letter · Decision Letter 0]

23 Oct 2019

Dear Dr Bachtrog,

Thank you very much for submitting your Research Article entitled 'Male recombination produced multiple geographically restricted neo-Y chromosome haplotypes of varying ages that correlate with onset of neo-Y decay in Drosophila albomicans' to PLOS Genetics. Your manuscript was fully evaluated at the editorial level and by independent peer reviewers. Please note that one of the reviewers (Reviewer #1) was the same as the last time we reviewed this paper, and the other two are different from the original version as those individuals were not available. The reviewers appreciated the attention to an important topic and also acknowledged the care you took to address the previous reviews. However, they also identified some aspects of the manuscript that should be improved.

We therefore ask you to modify the manuscript according to the review recommendations before we can consider your manuscript for acceptance. Your revisions should address the specific points made by each reviewer.

[LINK]

Yours sincerely,

Kelly A. Dyer

Associate Editor

PLOS Genetics

Bret Payseur

Section Editor: Evolution

PLOS Genetics

Reviewer's Responses to Questions

**Comments to the Authors:**

Reviewer #1: The authors have addressed most of my previous comments and they have done a considerable amount of additional work, including new tables and figures. I still have, however, some issues with the interpretation of the results shown in Table 2.

Table 2 (“both” column) shows that the fraction of genes (%) with either significant neo-X or neo-Y biased expression increases with the age of neo-Y haplotypes. However the next two columns show that the relative number of neo-X and neo-Y biased genes is maintained over time, which is difficult to explain under a simple model of gradual neo-Y degeneration. In my view, what we see is gene expression becoming more chromosome-specific (either neo-X or neo-Y biased but not both) with time. To me, this is perhaps an even more interesting observation. I feel that the manuscript would improve if this alternative interpretation were more clearly stated.

Suggested edits:

1. “Mai, Nalley and Bachtrog, in press” is the citation for the reference strain. Please include the full citation in the ‘References’ section.

2. “Together, these observations imply that the young neo-sex chromosomes of D. albomicans not only present a window to examine the initiating stages of sex chromosome differentiation”. “Initial” instead?

3. “Shorter branch lengths indicate that the neo-Y haplotype ceased to recombine with the neo-X sooner after the formation of the species, and, extensively, an older neo-Y haplotype”. Perhaps “shortly after species formation” instead?

4. “Monophyly of neo-Ys close to the centromere confirm that”. “confirms” instead?

5. “Based on the prevalent model whereby the neo-X and neo-Y fusions occurred after the species split [33-38], the two chromosomes are expected to be more similar to each other than to Chr. 3.” Which two chromosomes? Perhaps “...the two neo-sex chromosomes” instead?

6. “The distal end of Y3’s shorter D. nasuta haplotype block lands precisely at a haplotype breakpoint (Figure 3B and Figure 4D), arguing that male recombination truncated the segment with sequences from the neo-X.” The use of Y3’s is awkward. Perhaps “The distal end of the Y3 D. nasuta shorter haplotype” instead?

7. “Interestingly, no neo-X biased genes are shared across all three neo-Ys and no more ...”. Neo-Ys is a bit awkward. Perhaps “neo-Y groups”?

Reviewer #2: This study uses genome-wide polymorphism and expression data to investigate patterns of neo-Y chromosome variation across different populations of Drosophila albomicans. The authors present evidence of multiple geographically restricted neo-Y haplotypes. While the nucleotide variation within each haplotype is limited, ancestral recombination between the neo-X and the neo-Y chromosomes has led to high genetic diversity between the neo-Y haplotypes. In addition, as recombination in the different populations was lost at different time points, there is evidence of incipient Y decay, the extent of degeneration correlating with the age of the neo-Y haplotypes. Overall, I found this manuscript to be very thorough and very well written. I only have a few comments and queries.

In the manuscript section relating to the correlation between neo-Y degeneration and haplotype age (page 10), the authors report that the number of non-synonymous mutations increases with age. While the oldest Y3 haplotype does indeed have the most overall substitutions within CDS and the youngest Y2 haplotype has the fewest, the proportion of non-synonymous substitutions out of the total number of substitutions is actually the lowest for the Y3 haplotype and the highest for the Y2 haplotype (Table 3). How do the authors explain this pattern?

Page 4: I understand that neo-Y sites were identified from neo-X sites that were homozygous in the females but heterozygous in the males. However, in the section “This generated 255,010 variant neo-Y sites across the eleven neo-Ys and 584,333 across the neo-sex chromosomes after filtering.”, I am a bit confused about where the difference between these two numbers of variant neo-Y sites comes from.

Page 6: “disruption of monophyly of the neo-Ys are due to recombination events with neo-Xs” – I think here it should be “is due to” instead of “are due to”.

Page 20 Figure 1 legend: “neo-Y samples collected from each site is labeled, and

color-coded based on grouping in C” - Here “is labeled” should be replaced with “are labeled”.

Figure 6: In this plot, I think it might be useful to add labels for each neo-Y haplotype as for example on page 10 there is mention of significant differences between Y1 and Y3 in Figure 6 yet there are no labels in this figure to point to the different haplotypes.

There is missing information about the sex of the d.alb.SHL48 and d.alb.SHL48_M individuals in Supplementary Table 1.

Throughout the manuscript, the authors should ensure consistent use of either “p-value” or “P-value”; “neo-X biased” or “neo-X-biased”; “Chr. 3” or “Chr3”.

Reviewer #3: I upload the review as an attachment.

**Have all data underlying the figures and results presented in the manuscript been provided?**

Reviewer #1: Yes

Reviewer #2: None

Reviewer #3: Yes

PLOS authors have the option to publish the peer review history of their article (what does this mean?). If published, this will include your full peer review and any attached files.

Reviewer #1: No

Reviewer #2: No

Reviewer #3: No

---

## [Editor Report · Decision Letter 1]

1 Nov 2019

Dear Dr Bachtrog,

We are pleased to inform you that your manuscript entitled "Ancestral male recombination in Drosophila albomicans produced geographically restricted neo-Y chromosome haplotypes varying in age and onset of decay" has been editorially accepted for publication in PLOS Genetics. Congratulations!

Yours sincerely,

Kelly A. Dyer

Associate Editor

PLOS Genetics

Bret Payseur

Section Editor: Evolution

PLOS Genetics

Comments from the reviewers (if applicable):

**Data Deposition**

http://datadryad.org/submit?journalID=pgenetics&manu=PGENETICS-D-19-01486R1

**Press Queries**

---

## [Editor Report · Acceptance letter]

12 Nov 2019

PGENETICS-D-19-01486R1 

Ancestral male recombination in *Drosophila albomicans* produced geographically restricted neo-Y chromosome haplotypes varying in age and onset of decay 

Dear Dr Bachtrog, 

We are pleased to inform you that your manuscript entitled "Ancestral male recombination in *Drosophila albomicans* produced geographically restricted neo-Y chromosome haplotypes varying in age and onset of decay" has been formally accepted for publication in PLOS Genetics! Your manuscript is now with our production department and you will be notified of the publication date in due course.

With kind regards,

Matt Lyles

PLOS Genetics

On behalf of:
